# Progress on COVID-19 Chemotherapeutics Discovery and Novel Technology

**DOI:** 10.3390/molecules27238257

**Published:** 2022-11-26

**Authors:** Yalan Zhou, Huizhen Wang, Li Yang, Qingzhong Wang

**Affiliations:** Institute of Chinese Materia Medica, Shanghai University of Traditional Chinese Medicine, Shanghai 201203, China

**Keywords:** COVID-19, SARS-CoV-2, drug target, deep learning, CRISPR/Cas, multiomics analysis

## Abstract

COVID-19 is an infectious disease caused by severe acute respiratory syndrome coronavirus 2 (SARS-CoV-2), a novel highly contagious and pathogenic coronavirus that emerged in late 2019. SARS-CoV-2 spreads primarily through virus-containing droplets and small particles of air pollution, which greatly increases the risk of inhaling these virus particles when people are in close proximity. COVID-19 is spreading across the world, and the COVID-19 pandemic poses a threat to human health and public safety. To date, there are no specific vaccines or effective drugs against SARS-CoV-2. In this review, we focus on the enzyme targets of the virus and host that may be critical for the discovery of chemical compounds and natural products as antiviral drugs, and describe the development of potential antiviral drugs in the preclinical and clinical stages. At the same time, we summarize novel emerging technologies applied to the research on new drug development and the pathological mechanisms of COVID-19.

## 1. Introduction

COVID-19 is an infectious disease caused by severe acute respiratory syndrome coronavirus 2 (SARS-CoV-2) [1]. The most common symptoms include headache [2], loss of smell and taste [3,4], nasal congestion and runny nose, cough, muscle aches, sore throat, fever [2], diarrhea, and dyspnea. Although some infected individuals have no obvious symptoms, they may still transmit the virus, increasing the risk of disease transmission [5,6,7]. According to the World Health Organization (WHO), more than 600 million cases and 6.4 million deaths had been reported globally by 4 September 2022 [8]. At present, effective and preventive measures mainly include vaccination, staying at home, wearing masks in public places, maintaining physical or social distance, etc. [9]. Highly effective COVID-19 vaccines provide individuals with acquired immunity to the SARS-CoV-2 virus, which has been administered globally and slowed the spread of SARS-CoV-2 to some extent [10]. However, drug therapy is still very important for those individuals who are awaiting vaccination and those who have a strong response to vaccination. Therefore, the discovery of potential and effective drugs has important implications for the management and treatment of patients with COVID-19 [11]. Since the outbreak, scientists have been researching and developing effective drugs to treat COVID-19. To date, no drug has been approved to treat this severe disease [12]. In this review, we will focus on new drug discovery for COVID-19, as well as the latest research and analytical methods in the drug discovery process, to provide a theoretical basis for the further development of anti-COVID-19 drugs.

## 2. SARS-CoV-2 Virus and Pathogenic Mechanism

Coronaviruses are a large class of viruses that are widely distributed in nature. Coronaviruses have the largest genome among all viruses, and the expression mechanism of their genome in host cells is relatively complex. Both SARS-CoV-2 and SARS-CoV and MERS-CoV belong to the same category of beta coronaviruses and have very similar structures. SARS-CoV-2 has 82% genome homology with SARS-CoV [13]. SARS-CoV-2 is the seventh coronavirus that can infect humans [14]. When SARS-CoV-2 enters the host cell, the virus uses the host cell’s enzyme system, raw materials, and energy to replicate nucleic acids and translate the viral proteins using the host cell’s ribosome. The replication process can be divided into five steps: adsorption, penetration, decoating, biosynthesis, and release of the assembly [15]. In adsorption, the coronavirus binds to the specific protein receptor via the S protein on the cell surface and is adsorbed into the host cell [16]. Penetration is the entry of the viral nucleic acid or infectious nucleocapsid through the cell membrane into the cytoplasm, beginning the intracellular phase of viral infection of the host. A crucial step in the penetration process is the fusion of the viral envelope with the host cell membrane [17]. One possible mechanism could be that the virus is endocytosed by the host cell. After the coronavirus binds to the host cell receptor, the host cell membrane wraps around the virus, forms virus-containing endosomal vesicles in the form of endocytosis, and releases viral RNA into the infected cells. Another possible mechanism is the direct fusion of the lipid envelope of the virus with the host cell membrane and the release of viral RNA into the host cell cytoplasm [18,19,20]. The S-protein cleavage regions of coronaviruses can be recognized by specific sites, resulting in different membrane fusion efficiencies and thus affecting the replication efficiency of the virus. Recent studies have shown that SARS-CoV-2 has a potential furin-like protease cleavage site at the amino-terminus of the S1/S2 site and is processed during viral biosynthesis. This cleavage site is not present in SARS-CoV and could distinguish the S protein from SARS-CoV-2 and SARS-CoV. Based on the widespread expression of furin, it is speculated that the presence of this site may be involved in expanding the host tropism and tissue tropism of SARS-CoV-2, as well as increasing transmissibility and altering pathogenicity. After the viral infectious nucleic acid is released from the capsid, the subsequent synthesis of viral genomic RNA and subgenomic RNA is completed in the host cell. Subsequently, the genomic RNA and structural proteins are assembled, knotted into vesicles, and transported to the Golgi apparatus, where the TMPRSS2 S protein is cleaved to form an active new virus. After the virus has invaded, the human body produces an immune response to fight the virus. Highly pathogenic human coronaviruses can suppress the host’s innate and specific immune response by eliminating immune cells such as macrophages, monocytes, T lymphocytes, etc., which causes the human body to produce a large number of cytokines and chemokines [21,22,23,24]. This causes a cytokine storm, which leads to inflammation and edema at the site of the systemic lesion. The cytokine storm is an important cause of acute respiratory distress syndrome and multiple organ failure [4]. Cytokines over-secreted in SARS patients include IL-1β, IL-6, IL-12, gamma interferon, chemokine CXCL10, and monocytochemotactic protein-1 (MCP-1). Cytokines in patients include γ-interferon, TNF-α, IL-15, IL-17, IL-1β, IL-6, and IL-8 [5,6,7,8] and 1β, γ-interferon, inducible protein 10 (IP-10), and MCP-1 [9]. Compared with non-severe COVID-19 patients, the levels of IL-2, IL-7, IL-10, granulocyte-colony-stimulating factor, IP-10, and MCP-1 were significantly increased in severe patients [25]. This is the propagation and transmission mechanism of highly pathogenic human coronavirus and provides a basis for blocking its transmission and pathogenicity (Figure 1).

To date, there is no effective drug for the treatment of the SARS-CoV-2 virus [26]. In principle, all enzymes and proteins involved in viral replication and control of coronaviruses in host cells are potential drug targets [27]. Currently, the development of COVID-19 therapeutics mainly revolves around four nonstructural proteins (including the main coronavirus protease (Mpro, also known as 3C-likeprotease, 3CLpro), papain (PLpro), helicase, and RNA-dependent RNA polymerization enzyme (RdRP) [28]. These four nonstructural proteins are key enzymes in the virus life cycle. At the same time, the structural protein (S protein) is involved in the interaction between the virus and cell receptors during virus entry and is also an important anti-COVID-19 target for drug development [29,30,31] (Figure 2). We will discuss the above target proteins and recent drug discoveries.

## 3. Drugs That Target the 3CLpro

3C-like chymotrypsin (3C-likeprotease, 3CLpro, Mpro, nsp5), also known as M protease, is a functional protein required for viral replication [32,33,34,35]. This protease can hydrolyze at least 11 conserved sites in pp1a and pp1ab polyproteins, ensuring the production of nsp4~16 as important nonstructural proteins in viral transcription and replication, including RdRp, helicase, and Mpro itself. Previous studies have shown that Mpro proteins have high homology between different viral strains but no homology between host cells [36,37,38]. Therefore, Mpro is also considered one of the most important targets for antiviral drug development [39,40,41]. In October 2021, the oral M-protease inhibitor PF-07321332, developed by Pfizer, became the only small-molecule SARS-CoV-2 inhibitor candidate to enter phase III clinical trials. Similar to PF-07321332, most known M-protease inhibitors are mainly polypeptide inhibitors. Cys145 of Mpro forms an irreversible covalent bond through active groups, such as its own aldehyde group and α-ketoamide [25], thereby inactivating Mpro and exerting antiviral activity.

### 3.1. PF-07321332

PF-07321332 (Paxlovid, Nirmatrelvir, (Figure 2(**1**)) is a clinical drug candidate with potent viral inhibitory activity against SARS-CoV-2 Mpro [42,43,44,45,46]. Its enzyme inhibition constant Ki against SARS-CoV-2 Mpro in in vitro experiments is 3.11 nmol/L, its enzyme inhibitory activity is IC50 = 18.0 nmol/L, and its antiviral activity against SARS-CoV-2 virus infection is EC50 = 74~93 nmol/L. PF-07321332 has a comparatively low oral bioavailability, being demonstrated as 20–25% in pharmacokinetic experiments in rats and dogs. This is due to the fact that PF-07321332 itself is the metabolic substrate of the drug enzyme CYP3A4 in vivo [27]. For this reason, the strategy of combination with ritonavir, a potent CYP3A4 inhibitor (NCT04756531), has been used in clinical trials to improve the half-life and bioavailability of the drug in vivo [47,48]. The crystal structures of the drug and SARS-CoV-2 show that the CN substituent in the drug molecule binds covalently to the Cys145 residue in the protease, while this covalent binding mode was confirmed to be reversible by reversible SARS-CoV-2 inhibitor competition experiments [49,50]. A phase III clinical trial with PF-07321332 in patients with mild and moderate symptoms of COVID-19 was recently completed. The results show that the hospitalization rate in the placebo group decreased from 7 to 0.8% when this inhibitor was taken within 3 days of the onset of COVID-19 symptoms such as fever and dry cough, and the mortality rate decreased from 1.8 to 0% in 1219 adult patients who had not been vaccinated against COVID-19. This drug was submitted for emergency approval to the US FDA in November 2021.

### 3.2. Lopinavir/Ritonavir

*Lopinavir* (Figure 2(**3**)) and *ritonavir* (Figure 2(**2**)) are commonly used in the clinical treatment of patients infected with human immunodeficiency virus (HIV) in the form of fixed-dose combination drugs [51]. The manufacturing process of ritonavir involves, firstly, the synthesis of the formazan ox-carbonyl derivative fragment, the n-hexane fragment, and the valine derivative fragment, followed by amide condensation of the three fragments [52,53]. *Lopinavir* requires (2S,3S,5S)-2-amino, the succinate salt of 3-hydroxy-5-(tert-butoxycarbonylamino)-1,6-diphenylhexane as the starting material, which is prepared via a three-step reaction of amidation, amino deprotection, and amidation. One study showed that *lopinavir* and ritonavir can interact with the 3C-like chymotrypsin of MERS-CoV and SARS-CoV. The use of this combination drug improved symptoms in SARS-CoV patients in a nonrandomized open-label clinical trial [54]. Therefore, researchers are currently conducting a randomized, controlled, open-label clinical trial (ChiCTR2000029308) to evaluate the efficacy of *lopinavir* and ritonavir in hospitalized patients with severe COVID-19 [55,56,57].

### 3.3. GC376

GC376 (Figure 2(**4**)) is a protease inhibitor that blocks 3CLpro, a protease found in many (+)ssRNA viruses, thereby preventing the viral polyprotein from breaking down into its functional parts. Chemically, GC376 is the bisulfite adduct of the aldehyde GC373 and behaves like a prodrug for this compound [58]. GC376 was originally shown to be well-tolerated and effective in preventing feline coronavirus (FCoV) infection in animal studies [59,60]. Both GC373 (Figure 2(**5**)) and GC376 showed nanomolar inhibitory activity against SAR-CoV-2 cells (IC50 was 0.40 ± 0.05 and 0.19 ± 0.04 μmol/L, respectively). The co-crystal structure of GC376/GC373 and SARS-CoV-2 Mpro showed that the bisulfite functional group of GC376 was removed and Cys145 of Mpro formed a hemithioacetal with GC376/GC373 to inhibit the activity of Mpro [58]. Based on the vivo efficacy validated in animal models, this class of compounds is considered a promising candidate for the treatment of human coronavirus infections.

### 3.4. Baicalein

Baicalein (Figure 2(**6**)) is a rare non-covalent and non-peptoid small-molecule inhibitor among currently known inhibitors [61] and one of the main active ingredients of Shuanghuanglian oral liquid [62,63], which is a flavonoid compound. After Chinese researchers found that Shuanghuanglian preparations had a significant inhibitory effect on Mpro, they investigated three index components mentioned in the Chinese Pharmacopoeia (chlorogenic acid, forsythia, and baicalein) and another 25 compounds in Shuanghuanglian preparations. Nine compounds were identified as inhibitors of Mpro in the low micromolar range. Among them, researchers extensively characterized the binding of baicalein to Mpro, and the crystal structure of the complex of SARS-CoV-2 Mpro and baicalein was also determined. The studies showed that the inhibitory activity of baicalein against SARS-CoV-2 Mpro protease was IC50 = 0.94 ± 0.20 μmol/L, the antiviral replication activity of baicalein was EC50 = 2.94 ± 1.19 μmol/L, and the cytotoxicity was CC50 > 200 μmol/L. Crystal structure analysis showed that the specificity and binding mode of Cys145 baicalein, unlike those of peptidomimetic small-molecule inhibitors, effectively prevents the substrate from entering the active site by acting as a “shield” at the interface between the two catalytic duplex residues His41 and Cys145. This unique binding mode, together with the efficient binding of the ligand and the smaller molecular core, make baicalein an important lead compound for further development of Mpro inhibitors [64].

## 4. Drugs Targeting RNA-Dependent RNA Polymerase RdRp

The SARS-CoV-2 genome encodes a set of nonstructural proteins (nsp1~16) that, once inside the host cell, form the “core machinery” responsible for viral transcription and replication in a specific spatial and temporal order, forming the transcription–replication complex (RTC), which contains many important conservative antiviral targets. Additionally, the core component of the RTC is the RNA synthase nsp12 [65,66], which stimulates its enzymatic activity with the help of cofactors nsp7 and nsp8 and together with nsp7 and nsp8 forms a core RdRp complex [54]. RdRp is a type of RNA synthetase that uses viral RNA as a template and four nucleoside triphosphates (NTPs) as substrates to synthesize RNA [67,68,69,70,71]. Currently, there are more than 30 crystal structures of SARS-CoV-2 RNA synthase in various functional states in the RCSB protein database [72].

### 4.1. Molnupiravir

Molnupiravir (MK-4482, EIDD-2801, (Figure 2(**7**)), which has been classified as a mutagenic nucleotide analog, is an oral RdRp inhibitor for the treatment of mild-to-moderate new coronavirus infections in adults [73,74] and was the first drug to be approved for clinical use by the Medicines and Healthcare products Regulatory Agency (MHRA) in the United Kingdom and the first to be approved by the US FDA. It has been shown in clinical trials to be an ideal treatment for patients with COVID-19, both in terms of efficacy and safety [75,76,77,78]. Therefore, molnupiravir was selected as a clinical drug candidate [79,80]. In investigating the pharmacodynamic mechanism, researchers found that molnupiravir can effectively insert into the RNA template of the SARS-CoV-2 virus, which promotes mutation and termination of viral replication [81]. In March 2021, Merck’s phase II clinical results showed that a positive viral detection rate of 0% was achieved in all dose groups (placebo) in patients with new-onset coronary pneumonia who received the drug again from day 5 [82]. The most recent phase III clinical results show that the risk of death and hospitalization was reduced by 50% in patients treated with molnupiravir. The probability of hospitalization or death was 7.3% in patients treated with molnupiravir on day 29 after randomization. In terms of safety, approximately 1.3% of patients in the molnupiravir-treated group discontinued treatment due to drug-related adverse events, and the rates of adverse effects of any magnitude were 35 and 40% in the treatment and placebo groups, respectively [74].

### 4.2. Favipiravir

Favipiravir (T-705, (Figure 2(**8**)) is a guanine analog used to treat influenza. It competitively inhibits entry into infected cells by simulating guanosine triphosphate (GTP) insertion into the extended RNA chain RdRp, thereby inhibiting viral genome replication and transcription and providing viral anti-RNA activity [83,84]. Due to its beneficial in vitro inhibitory activity, favipiravir was the first drug to receive “emergency approval” for the treatment of new coronary pneumonia in India in May 2020, and the Russian health authority also granted the drug emergency approval for the treatment of new coronary pneumonia [85]. Favipiravir has completed a phase III clinical trial for critically ill patients with new coronary pneumonia in Japan and is currently in the “pre-registration” phase [86]. However, as the results of the drug in the clinical phase are still controversial, Peng et al. (2021) confirmed the cryo-electron microscopic structure of the RdRp complex of favipiravir and SARS-CoV-2 in the pre-catalyzed state to better understand the antiviral mechanism of the drug. The study found that the drug cannot be used if it is not present in the RNA template. To repeat the pyrimidine residues, the presence of favipiravir in the RNA product does not require immediate termination of the extension of the growing chain, and favipiravir is able to escape the proofreading mechanism of the replication and transcription complex (RTC), resulting in mutation of the viral genome of the progeny and thereby exerting antiviral effects [87,88]. This result also shows that the drug does not significantly inhibit RNA synthesis in vitro.

### 4.3. Remdesivir

Remdesivir (GS-5734, (Figure 2(**9**)) is a prodrug of nucleoside analogs explored by Gilead Sciences that can be synthesized from ribose derivatives by multistep reactions [89]. Remdesivir can inhibit SARS-CoV-2 in vitro assays [90], with a mechanism of action similar to that of favipiravir. However, clinical studies of its efficacy have yielded conflicting results. Based on the results of an adaptive clinical trial for the treatment of COVID-19, the drug was approved by the US FDA for the treatment of hospitalized patients with COVID-19 [91]. The World Health Organization believes that the drug may improve patient survival or reduce patients’ need for oxygen inhalation without significant side effects based on a “robust clinical trial” [92]. Recently, Yin et al. (2020) [93] and Bravo et al. (2021) [94] obtained the cryo-electron microscopic crystal structure of the drug and target protein RdRp, which can help scientists understand the antiviral mechanism of remdesivir and guide the rational design of improved antiviral drugs. These electron microscopic structures show that the template RNA of motif F and motif G can be recognized by nsp12, and the chain is then inserted into the core site, thereby executing the chain extension mechanism to replicate the viral genome. Studies have shown that the antiviral mechanism of remdesivir is due to the cyano group (CN) contained in the molecular structure of remdesivir colliding with Ser861 on nsp12 to block RNA translocation after the single phosphate ester of remdesivir is incorporated into the base, resulting in chain termination [94].

## 5. Drugs That Target Spike Glycoproteins

The spike glycoprotein (S protein, S, spike protein) of the new coronavirus is a structural protein consisting of 1200–1500 amino acids and containing 21–35 N-glycosylation sites. Several S proteins form a special spike-shaped corolla structure on the surface of the coronavirus in the form of a trimer [95,96]. The main function of the S protein is to bind with angiotensin-converting enzyme 2 (ACE2 enzyme) on the surface of human cells so that coronavirus particles can be introduced into the cell and replicated to produce more next–generationvirusparticles [97,98]. TheSproteiniscleavedintotwosubunitsunderthe action of host cell proteases S1 and S2. S1 is responsible for cell recognition by ACE2, while S2 mediates membrane fusion between the virus and the host [99]. Therefore, the S protein is a very important target for prevention and control of infection and dissemination of coronavirus drugs. However, the new coronaviruses currently spreading in countries around the world are all mutated at the S protein, which may increase the transmissibility of the virus, make the virus more harmful to the host, or decrease the ability of vaccine-induced antibody neutralization [100,101,102,103,104,105,106,107,108,109]. Additionally, these mutations also make the development of antiviral drugs targeting the S protein an extremely difficult task. It is worth noting that the S proteins of SARS-CoV-1 and SARS-CoV-2 share 76.9% sequence identity [110], which means that drugs developed against the S protein of SARS-CoV-1 can also target the S protein of SARS-CoV-2.

### 5.1. Arbidol

Arbidol (Figure 2(**10**)) is an antiviral drug developed by the Soviet Medicinal Chemistry Research Center that can prevent the viral envelope from contacting, adhering to, and fusing with the host cell membrane [111,112]. The results of molecular dynamics simulation analysis show that arbidol with a high-affinity combination is able to stabilize at the interface between the receptor-binding domain (RBD) and ACE2. However, the crystal structure of the drug and SARS-CoV-2 has not been successfully analyzed [113]. Arbidol inhibits a variety of viruses, including influenza virus, respiratory syncytial virus (RSV), atypical pneumonia virus (SARS-CoV), hepatitis C virus (HCV), and hepatitis B virus (HBV) [114]. Recent studies have shown that the arbidol treatment group did not achieve better clinical efficacy than the control group with lopinavir/ritonavir combination therapy in terms of accelerating viral clearance and improving clinical symptoms. Therefore, the therapeutic effect of this drug on COVID-19 needs further investigation [115].

### 5.2. SSAA09E2

SSAA09E2 (Figure 2(**11**)) is an oxazole carboxamide derivative that Adedeji et al. (2013) studied by assembling a library of 3000 compounds according to Lipinski’s five rules [116], which can block the binding of the RBD of the SARS-CoV S protein to ACE2 with an IC50 value of 3.1 μmol/L and a CC50 of >100 μmol/L. Further studies confirmed that SSAA09E2 did not affect ACE2 expression but likely impaired recognition of the ACE2 protein by the RBD domain in the S1 subunit of the virus [116,117,118].

## 6. Novel Technologies Applications in COVID-19 Drug Screening

SARS-CoV-2 has been constantly mutating and has produced a variety of variants (also known as mutant strains). Mutant strains that are closely monitored by the World Health Organization (WHO) include alpha, beta, gamma, delta, and omicron [119]. The current widespread Omicron variant has more than 30 mutated sites in its surface spike protein (also known as S protein), many of which are already present in the alpha and delta variants and are related to the infectivity of the virus and its ability to evade neutralization by antibodies [120]. The constant emergence of new mutations in the structure of the virus has posed significant challenges to drug discovery and development. Screening potential drugs through basic experiments is not sufficient to deal with the rapid spread of the new coronavirus. The application of novel technologies can effectively shorten the time and cost of drug development, which is of far-reaching importance for the treatment of COVID-19.

### 6.1. Deep Learning Technology

In the early stages of the epidemic, clinical researchers conducted a large number of repeat studies because no specific drugs were available. Therefore, deep learning has begun to be used to assist in the development of new drugs, the degradation of existing drugs, and the development of vaccines in the absence of clinically proven drugs and vaccines [121,122,123,124]. Considering the current shortage of agents against SARS-CoV-2, the solutions provided by deep learning can accelerate the development and optimization of specific drugs/new drugs. Deep learning has already achieved some results in drug screening for COVID-19; for example, Gysi et al. (2021) [125] constructed a network medicine framework for drug reuse by combining AI, network propagation, and network distance algorithms, and derived the correlation between the two diseases by comparing the distance between the disease module COVID-19 and the other 299 disease modules. The drug repositioning approach not only predicts 6 potential drug candidates out of 918 drugs, but also proposes an algorithmic toolset with which to rapidly identify treatments to fill the gaps in disease treatment during traditional new drug development. Ghosh et al. (2021) [126] found that heterocyclic nuclei such as oxadiazole, furan, and pyridine appear to have positive effects on Mpro inhibition, while thiophene, thiazole, and pyrimidine may have negative effects. Deep learning was used to discover active compounds against SARS-CoV-2 Mpro from 1.3 billion compounds [127]. Wang et al. (2021) found a compound that they identified as a 3C-like protease inhibitor of SARS-CoV-2 after screening 4.9 million drug-like molecules from the ZINC15 database with a Covidv-3 dl-based model [128]. Deep learning identified baricitinib, atasanavir, and other hepatitis C antiviral drugs as effective drugs against COVID-19 [129,130,131]. In addition to the above compounds, Zhang et al. (2020) also found 26 herbs containing components against COVID-19 through molecular docking and network pharmacological analysis [132].

### 6.2. CRISPR/Cas13 Technology

In the last two decades, several variants of coronavirus, such as SARS-CoV-2, SARS, and MERS, have emerged from animal sources and infected humans, each time resulting in remarkable rates of illness and death. Therefore, there was an urgent need to develop a strategy to largely contain and stop the viruses, especially all the coronavirus strains currently found in animals. CRISPR/Cas13 has proven to be a robust approach to suppress gene expression in eukaryotes at the post-transcriptional level by altering encoded information in the genome at the RNA level. It is a class 2 ribonuclease of the type VI CRISPR-associated RNA-targeted ribonuclease. Four protein families are known, including Cas13a (originally designated as class 2 candidate 2 (C2c2)), Cas13b, Cas13c, and Cas13d (named CasRx) [133,134,135]. The genome-editing capabilities of the CRISPR-CAS system have been widely recognized, which has sparked interest in developing new biosensing applications for nucleic acid detection. Recently, the method was developed for the detection of SARS-CoV-2, which has made significant progress. For example, Wahab A Khan et al. (2021) used the CRISPR-cas13a system to diagnose 60 COVID-19 patient samples with 100% detection accuracy. This approach highlights new ideas for infectious disease identification and can be extended to measure nucleic acids from other clinical isolates [136]. Abbott et al. (2020) altered the RNA-directed RNA endonuclease activity of Cas13d in human cells against SARS-CoV-2 and IAV virus targets. They demonstrated that the Cas13d system can effectively target and lyse the RNA sequence of the SARS-CoV-2 fragment and IAV in lung epithelial cells. In addition, their bioinformatic analysis showed that the smallest six crRNA pools can target 92% of the 91,600 IAV strains and the six crRNAs can target 91% of the 3051 sequenced coronaviruses. This extends the use of the CRISPR-Cas13 system beyond diagnostics [137]. Xu et al. (2021) used the smallest Cas protein, Cas13x, for bioinformatics analysis and selected crRNAs targeting the RNA sites of SARS-CoV-2RdRP and E proteins to test the antiviral ability of Cas13x. After co-transfection of the viral reporter gene with Cas13x/crRNA in HEK293T cells, they observed that the degradation efficiency of the viral RNA could reach 70%, indicating that Cas13x should be an effective agent against SARS-CoV-2 [138].

### 6.3. Multiomics Methods

With the continuous development of new technologies, it is possible to obtain high-dimensional tissue data quickly and efficiently. A disease is often influenced by multiple genetic variants in which DNA, RNA, proteins, and metabolites often play a role. Therefore, comprehensive analysis of multiomics data is of great importance for the study of complex biological processes and disease mechanisms [139]. Multiomics analysis refers to the normalization and comparative analysis of data sources from different omics, the establishment of the relationship between different groups, and the comprehensive multiomics data to comprehensively and deeply interpret biological processes at the gene, transcription, protein, and metabolic levels to better understand the biological system. The development of the multiomics method has enriched the understanding of the etiology and infection mechanism of SARS-CoV-2 and provided a theoretical basis for the new use of old drugs in clinical practice. More importantly, it has promoted the development of new drugs and the exploration of therapeutic targets for vaccines. By detecting and validating extensive information from biological samples, multiomics has played an important role in screening targets for the treatment of COVID-19, guiding the development of drug candidates, and accelerating the prevention of SARS-CoV-2 and other coronaviruses [140]. For example, Emily Stephenson et al. (2021) performed single-cell transcriptome, surface proteome, and T- and B-lymphocyte antigen receptor analyses on more than 780,000 peripheral blood mononuclear cells from 130 COVID-19 patients with varying severity and found the expression of complement transcripts from non-classical monocytes to sequester platelets and replenish the alveolar macrophage pool in COVID-19. Their research reveals a synergistic immune response contributing to the pathogenesis of COVID-19 and uncovers individual cellular components that are candidates for therapy [141]. In a recent multiomics study by Overmyer et al. (2020) [142], transcripts, proteins, metabolites, and lipids were quantified in patients with COVID-19 and COVID-19-like symptoms. These molecules were then associated with clinical outcomes, including concomitant disease, ICU status, and disease severity, using correlation analyses and machine learning techniques. From these analyses, the unique features of the disease were found to be lipid transport system dysregulation, complement activation, and neutrophil activation. Further enrichment analyses revealed that important signaling pathways of COVID-19 were summarized by their first two principal components, which were then used as predictors to assess the importance of these pathways for COVID-19 status and disease severity [143]. Gordon et al. (2020) identified 332 high-reliability interactions between viruses and human proteins using affinity purification mass spectrometry, and on this basis identified 66 human proteins and 69 protein-targeting compounds as potential targets [144]. Suet et al. (2020), through a comprehensive analysis of multiple omics datasets, found that there is significant immune transfer between mild and moderate COVID-19 infection and that in moderately severe cases, new proliferation and depletion of CD8+ T-cell subsets increases with the increase in disease severity. The functional S100highHLA-DRlow monocyte subsets identified by multiomics in plasma were also associated with the severity of COVID-19 [145].

## 7. Outlook

Rapid development of effective treatments for COVID-19 is a major challenge. Currently, scientists have successfully analyzed the crystal structure of most SARS-CoV-2 target proteins, such as spike protein, major protein, RNA-dependent RNA polymerase, etc. This review mainly described the pathogenic process of SARS-CoV-2 infection in humans, summarized some agents against COVID-19, and reviewed some new technologies that may be applied to the research on COVID-19. Considering that COVID-19 frequently mutates at the genetic level and escapes from the immune system, the application of new technologies could improve mitigation of the spread of the epidemic and reduce the time and cost of drug discovery and development. Recently emerged technologies, such as deep learning and multiomics, will greatly accelerate the screening of promising hit compounds against COVID-19. For example, baricitinib, atasanavir, and other antiviral drugs have been identified by deep learning as effective drugs against COVID-19 [129,130,131]; for the genome-editing technology, Abbott et al. (2020) have shown that the Cas13d system can effectively target and lyse SARS-CoV-2 fragments and IAV RNA sequences in lung epithelial cells. In addition, it has also been reported that the smallest six crRNA pools can target 92% of 91,600 IAV strains and six crRNAs targeted 91% of 3051 sequenced coronaviruses, extending the use of the CRISPR-Cas13 system beyond diagnosis [137]. Affinity purification mass spectrometry has also been used to identify 332 highly reliable interactions between viruses and human proteins, and on this basis 66 human proteins and 69 protein-targeting compounds have been identified as potential targets [144]. These studies have shown that new technologies play an indispensable role in the investigation of the mechanisms of COVID-19 and novel treatment strategies. Additionally, the combination of traditional Chinese and Western medicine can significantly alleviate symptoms of COVID-19. Lin Liping et al. (2022) found that the combination of sinomenium and remdesivir had a significant synergistic inhibitory effect on COVID-19 at the cellular level [146,147,148].

In summary, multi-dimensional strategies, including novel technologies, TCM and Western medicines, and nucleotide vaccines, will greatly improve the prevention of the spread of COVID-19.

## Figures and Tables

**Figure 1 molecules-27-08257-f001:**
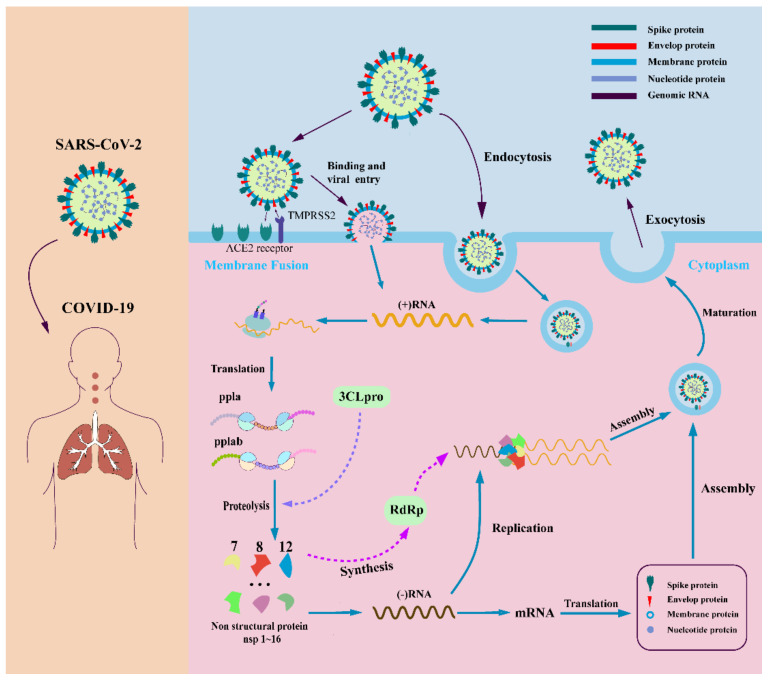
SARS-CoV-2 infection mechanism. SARS-CoV-2 enters the host cell by binding to specific host cell receptors and becoming endocytosed. The materials in the host cell are used to produce materials needed for viral replication and assembly by viral genes. After maturation of the virus, release into the extracellular space occurs.

**Figure 2 molecules-27-08257-f002:**
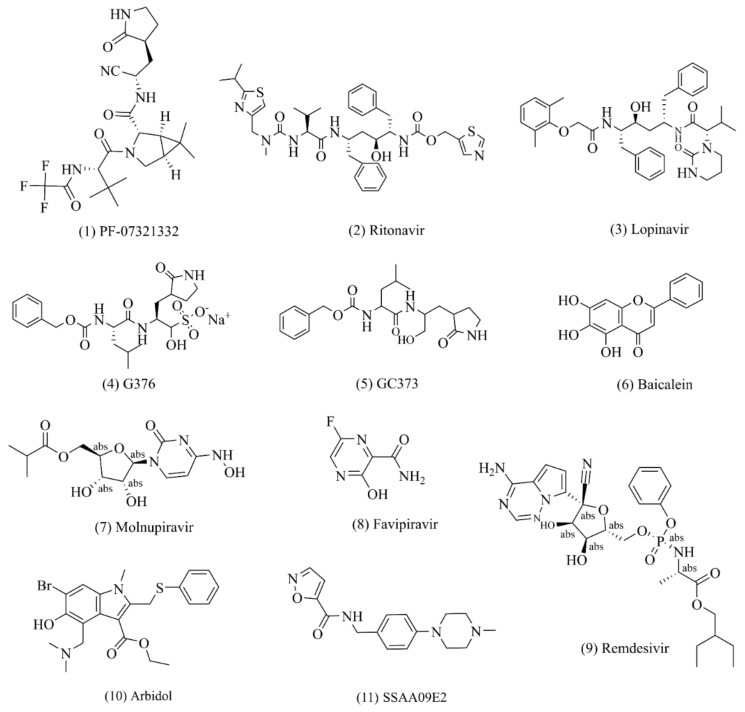
Potential drugs against SARS-CoV-2. (**1**)–(**6**): small drug molecules acting on 3CLpro targets; (**7**)–(**9**): small drug molecules acting on RdRp targets; (**10**)–(**11**): small drug molecules acting on spike protein.

## Data Availability

Not applicable.

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
