# Peer review of "Progress on COVID-19 Chemotherapeutics Discovery and Novel Technology"

_molecules, 2022, doi:10.3390/molecules27238257_

Round 1
Reviewer 1 Report
REVIEWER'S REPORT
Manucsript title: Progress on COVID-19 Chemotherapeutics Discovery and 2 Novel Technology (Authors: Yalan Zhou , Huizhen Wang, Li Yang , Qingzhong Wang )
This review-article concentrates on the identification for COVID-19, most recent research and analytical approaches in the drug discovery process, in order to provide a theoretical fundation for future development of anti-COVID-19 therapies, despite the fact that no medications has yet been approved to treat this desease. This article, in my opinion, will likely to catch the interest of readers who work in this or related subjects, and it may be published in this journal with modest editing and supplemention.
Albeit the manuscript is written in quite acceptable English, I would still advise proofreading the entire text as it may substantially enhance the text quality
Minor remarks:
Enzyme Complex Crystal Structures in Table 1 (pages 4-6, line 106) gives the impression that these structures are only formally tossed into the table due to the general background. So, what sense of that? What information does the reader gain by looking at these structures? In general, I would advise structuring the most crucial information (for each medicine covered in this article) in Table 1. It would be much more easier for the reader to grasp and recall the most important details of the compound's activity against COVID-19.
In page 21, line 335-336, "The drug reuse method...." I recommend to change to "The drug repositioning ( reprofiling or re-tasking) approach (or technique)....."
In my opinion, the "Outlook" subsection (pages 21-22 ) is described in a fairly formal manner. Given the context of this manuscript, I would recommend better aranging this subsection so that reader can perceive the emphasis on crucial points.
Reviewer 2 Report
This reviewer has the following suggestions regarding the review paper submitted to Pharmaceutics;
1. COVID-19 is no more new world and may be stated without its full form.
2. Improve the text within Figure 1.
3. 12 compounds in Table 1 are too fine and it is difficult to read them. They may be improved to become prominent. I opine to delete the Table, gather all 12 compounds in a Figure, number them from 1-12 and then elaborate them in the text. As has already been done for some of the compounds in the following text. Trade name, chemical name and all other properties shall be discussed in the text.
4. Provide a visual for Technologies being in practice, such as Deep learning, CRISPR-Cas13 technology (CRISPR/Cas13 check and unify too) and Multi-omics methods. Some statistical data may be collected to highlight the importance of the method and their applicability.
5. Some numerical values for discussion is needed to further strengthen the review.
6. In its present form the balance between text and List of references does not look nice. Keeping in view the balance between various sections it is suggested to elaborate the discussion in more critical way.
7. The list of 165 recent references is enough but a clear trend needs to be developed from these references.
Round 2
Reviewer 2 Report
The authors have made all the suggested changes to the entire satisfaction of this reviewer. Still I suggest to enhance the font size of atoms in Chemdraw structures of compounds, make all compounds uniform (look at ring sizes, some are smaller than other). Look at the Fig 2, and improve it. Bring the molecules closer to each other at optimum distance to fill the unnecessary space.
The modification may be carried out in final version, I do not feel hesitation to favor publication of the review article in Molecules.
